# A Positive Causal Effect of Shrimp Allergy on Major Depressive Disorder Mediated by Allergy- and Immune-Related Pathways in the East Asian Population

**DOI:** 10.3390/nu16010079

**Published:** 2023-12-26

**Authors:** Shitao Rao, Xiaotong Chen, Olivia Yanlai Ou, Sek Ying Chair, Wai Tong Chien, Guangming Liu, Mary Miu Yee Waye

**Affiliations:** 1Department of Bioinformatics, Fujian Key Laboratory of Medical Bioinformatics, Institute of Precision Medicine, School of Medical Technology and Engineering, Fujian Medical University, Fuzhou 350122, China; strao@fjmu.edu.cn (S.R.); chenxt@fjmu.edu.cn (X.C.); 2School of Biomedical Sciences, The Chinese University of Hong Kong, Hong Kong, China; 3Department of Psychology, University of Toronto, Toronto, ON M5S 1A1, Canada; olivia.ou@mail.utoronto.ca; 4Croucher Laboratory for Human Genomics, Asia-Pacific Genomic and Genetic Nursing Centre, The Nethersole School of Nursing, The Chinese University of Hong Kong, Hong Kong, China; sychair@cuhk.edu.hk (S.Y.C.); wtchien@cuhk.edu.hk (W.T.C.); 5College of Ocean Food and Biological Engineering, Xiamen Key Laboratory of Marine Functional Food, Fujian Provincial Engineering Technology Research Center of Marine Functional Food, Jimei University, Xiamen 361021, China

**Keywords:** shrimp allergy, major depressive disorder, East Asian, Mendelian randomization, pleiotropic genes, shared genetic basis

## Abstract

Background: Observational studies have implied a potential correlation between allergic diseases and major depressive disorder (MDD). However, the relationship is still inconclusive as it is likely to be interfered with by substantial confounding factors and potential reverse causality. The present study aimed to investigate causal correlation of the two diseases by a Mendelian randomization (MR) study and further elucidate the underlying molecular mechanisms. Methods: With the biggest summary datasets of a genome-wide association study (GWAS) in the East Asian population, we conducted a two-sample, bidirectional MR study to assess the causal correlation between shrimp allergy (SA) and MDD. Subsequently, we identified the pleiotropic genes’ susceptibility to the two diseases at whole-genome and tissue-specific levels, respectively. Enriched GO sets and KEGG pathways were also discovered to elucidate the potential underlying mechanisms. Results: With the most suitable MR method, SA was identified as a causal risk factor for MDD based on three different groups of independent genetic instruments, respectively (*p* < 2.81 × 10^−2^). In contrast, we did not observe a significant causal effect of MDD on SA. The GWAS-pairwise program successfully identified seven pleiotropic genetic variants (PPA3 > 0.8), indicating that the two diseases indeed have a shared genetic basis. At a whole-genome level, the MAGMA program identified 44 pleiotropic genes, which were enriched in allergy-related pathways, such as antigen processing and presentation pathway (*p* = 1.46 × 10^−2^). In brain-specific tissue, the S-MultiXcan program found 17 pleiotropic genes that were significantly enriched in immune-related pathways and GO sets, including asthma-related pathway, T-cell activation-related, and major histocompatibility complex protein-related GO sets. Regarding whole-blood tissue, the program identified six pleiotropic genes that are significantly enriched in tolerance induction-related GO sets. Conclusions: The present study for the first time indicated a significant causal effect of SA on the occurrence of MDD, but the reverse was not true. Enrichment analyses of pleiotropic genes at whole-genome and tissue-specific levels implied the involvement of allergy and immune-related pathways in the shared genetic mechanism of the two diseases. Elucidating the causal effect and the acting direction may be beneficial in reducing the incidence rate of MDD for the massive group of SA patients in the East Asian region.

## 1. Introduction

In recent years, food allergy (FA) has become a widely recognized and highly prioritized food safety issue for governments and the general public worldwide, particularly in China where the incidence rate continues to rise [1,2]. Food allergy refers to type I hypersensitivity reactions mediated by immunoglobulin E (IgE), which happen rapidly and manifest typically as serious damage in skin barrier, dysfunction of respiratory and central nervous systems, and even death in severe cases [3]. Patients with allergic diseases are often reported to suffer from depression and vice versa [4]. However, the actual relationship of the two diseases has remained inconclusive. The major depressive disorder (MDD) is a type of common mood disorder in the central nervous system and usually causes severe patients to have suicidal thoughts and further take fatal action through suicidal attempts, which brings a heavy mental and economic burden to the patient’s family and the entire community [5,6].

The main eight categories of allergenic foods stipulated by the World Health Organization/Food and Agriculture Organization include eggs, milk, peanuts, soybeans, nuts, wheat, fish, and crustaceans [7]. In Western countries, food allergy reactions are mostly caused by peanuts, nuts, and milk [8], while allergies to seafood such as shrimp and fish are more common in the Asian region [9]. A recent study found that approximately 33% of allergic adults and 40% of allergic children in Asia were allergic to crustacean shellfish, mainly including shrimp and crab [10]. A number of recent observational studies suggested a potential correlation between allergic diseases and MDD [4,11,12]. Several previous studies have used statistics and found a high incidence of allergic diseases in spring and winter [13,14,15]. Meanwhile, the high prevalence of MDD and the suicide mortality it causes also happen in spring, indicating that the peak periods of the two diseases are highly coincident. A national cohort study in Korea reported a significant bidirectional association between asthma and depression in the form of a retrospective study [4]. However, correlations from observational studies may be substantially biased by many confounding factors and a potential reverse causality [16,17]. In addition, there is a lack of solid correlation studies for MDD and allergy to seafood, especially for the main shrimp allergy (SA) in the Asian population. Taken together, a more reliable method is urgently required to elucidate the correlation between MDD and the main SA for Asian people.

A Mendelian randomization (MR) study utilized dozens to thousands of independent single nucleotide polymorphisms (SNPs) as instrumental variables to infer causal correlations between exposure and outcome traits. As alleles in genetic variants are randomly distributed, the principle of MR could be considered similar to a randomized controlled trial (RCT) and is able to overcome the inherent issues in observational studies [17,18]. A large amount of genetic research has confirmed that FA has a very high heritability of about 80%, manifested through the fact that the incidence of FA in the same family is significantly higher than the average rate [19,20,21]. The heritability of MDD is moderate, with an estimated heritability ranging from 33 to 50% [22,23]. With the increase of summary statistics from a genome-wide association study (GWAS) with a large sample size, the MR method is a robust way to infer causality between two traits with a moderate to high heritability [24,25,26,27,28,29].

In the present study, we applied a two-sample, bidirectional MR study to assess the causal correlations between SA and MDD with the biggest summary datasets of GWAS in the East Asian population. Subsequently, we employed the GWAS-PW program to investigate if there is a shared genetic basis in the genetic etiology of the two diseases. Upon obtaining positive findings, we further utilized advanced bioinformatics tools to identify their pleiotropic genes and pathways at a whole-genome level and at a tissue-specific level, respectively. The shared genetic basis and potential pleiotropic genes/pathways could partially explain the underlying molecular mechanisms. Elucidating the acting direction of causal effect may be beneficial in reducing the incidence rate of MDD for the large group of SA patients in the East Asian region.

## 2. Materials and Methods

### 2.1. GWAS Summary Statistics

We collected the biggest summary datasets of genome-wide association studies (GWASs) for shrimp allergy (SA) and major depressive disorder (MDD) from published studies and publicly available databases. Notably, both of the two GWAS summary datasets were based on subjects of East Asian ancestry and were originally subjected to proper correction regarding population stratification to avoid the negative effect of slightly different genetic backgrounds. Details of the two datasets are shown in Table 1 and references therein.

The GWAS summary statistics of SA were conducted for self-reported reactivity with shrimp from approximately 8900 Japanese women (539 cases and 8350 controls) [30]. SA cases were defined as those that provided an affirmative response in either severe allergic reactions, namely ‘consciousness disorder, anaphylactic shock, such as a drop in blood pressure’, or mild allergic reactions, namely ‘itching, hives, swelling of the lips and eyelids, vomiting’. The selected samples were genotyped by Takara Bio (Kusatsu, Japan) on a custom East Asian-specific Axiom array (EverGene1), in which most variants have a minor allele frequency (MAF) above 0.01 in the 1000 Genomes Project Japanese ancestry samples. This published study was approved by the Institutional Review Board at the Tsukuba International Clinical Pharmacology Clinic.

The GWAS summary statistics of MDD were from a meta-analysis dataset, including the China, Oxford, and Virginia Commonwealth University Experimental Research on Genetic Epidemiology (CONVERGE) consortium, China Kadoorie Biobank (CKB), and the Taiwan-MDD study, as well as studies conducted in the USA and UK, with 194,548 individuals in East Asian ancestry (15,771 cases and 78,777 controls) [31]. MDD cases were defined by a range of measures including structured clinical interviews, medical health care records, symptom questionnaires, and self-completed surveys. Details of these cohorts and their genotyping methods in different studies were described separately in Giannakopoulou’s meta-study [31]. For the meta-analysis, variants presented in at least 2 studies were selected to perform a weighted z-score meta-analysis using the METAL program (version 25/03/2011) [32].

### 2.2. Two-Sample MR Study Design and Analysis

The primary MR analyses mainly contained one set of bidirectional analyses. On one side, we treated SA as an exposure and MDD as an outcome to evaluate whether the SA would genetically affect the incidence rate of MDD (Figure 1A,B). Conversely, we considered MDD as an exposure and SA as an outcome to identify if there is any genetically causal effect of MDD on the occurrence rate of SA (Figure 1A,C). Notably, despite slight differences in the populations for SA and MDD, they are commonly categorized within the East Asian population (EAS). This population subset is considered an integral component of the five super-populations outlined in the 1000 Genomes reference panel. Consequently, it aligns with the assumptions underlying the Mendelian randomization (MR) approach [33,34,35].

Both GWAS summary datasets went through multiple pre-processing steps. First, a missing SNP ID was added, and duplicate SNPs were deleted based on the reference genome (GRCh37). The summary dataset of MDD contains incomplete non-effect alleles from a meta-analysis of East Asian individuals with 9 cohorts, which would be added based on the GRCh37 reference genome as well. After adding the missing parameters, we further removed SNPs with a strong genetic correlation in the linkage disequilibrium block (*r*^2^ < 0.1, distance = 1000 kb) using the PLINK program (version 1.90). The correlations between SNPs were derived from the 1000 Genomes East Asian samples.

For the valid and independent SNPs, we first performed F-statistic testing to access their effects as instrumental variables (IVs) with the formula shown below [36,37]:$$F=\frac{{beta}^{2}}{{se}^{2}}$$

Those independent SNPs with an original *p*-value below the suggestive threshold of GWAS (5 × 10^−6^) were selected as Group 1. After F-statistic testing, all SNPs in group 1 were identified as strong IVs (F-statistic > 10). Meanwhile, all SNPs with an F-statistic greater than 10 were defined as strong IVs and selected for follow-up MR analyses (Group 2). In addition, we employed a more relaxed *p*-value threshold (5 × 10^−3^) to select genetic variants (Group 3) for further validating the significant genetic correlations found in the former two groups of SNPs.

All two-sample MR analyses were performed using the TwoSampleMR (version 0.4.26) package in the R programming language (version 4.2.0). Unlike common applications, we applied the most suitable MR approach in different scenarios. We first employed the MR Egger regression method to determine if there was a horizontal pleiotropy for the genetic instruments and the MR inverse-variance weighted (IVW) method to evaluate if there was an obvious heterogeneity among multiple genetic instruments. Then, the most suitable MR approach was determined in four different scenarios. The fixed-effect IVW method was applied when there was neither horizontal pleiotropy nor significant heterogeneity (scenario 1, S1). The random-effect IVW method was employed when there was obvious heterogeneity but no horizontal pleiotropy (S2). The MR Egger method or weighted median was applied when there was an obvious horizontal pleiotropy (S3). The Wald ratio method was employed when there was only one valid genetic instrument in MR analysis (S4). With that, only one hypothesis was tested for each pair of exposure and outcome, from which a *p*-value below 0.05 was considered indicative of significant genetic correlations. In terms of effect size in causal correlation when the exposure was a binary trait, the odds ratio (OR) could be roughly considered to represent the likelihood that the outcome would occur when an individual is exposed to one specific condition. In addition, we employed the Steiger directionality test to evaluate if the causal directions between the hypothesized exposures and outcomes are true [38].

### 2.3. Identification of a Shared Genetic Basis between SA and MDD

To discover the potential shared genetic basis between SA and MDD, we utilized three different approaches with different principles (GWAS-PW, MAGMA, and MetaXcan) to identify pleiotropic genes from the two summary datasets of GWAS.

The GWAS-PW program was designed to estimate the probability that a variant influences two traits at the same time [39]. In brief, the Bayesian method in this program uses summary statistics from one pair of traits to calculate the posterior probabilities of association (PPA) that an SNP is specifically influencing neither trait (models 0), only 1 of the 2 traits (models 1 and 2), or jointly both traits (model 3). Genomic loci with PPA ≥ 0.8 in model 3 were considered to have a pleiotropic effect on both SA and MDD. Subsequently, these pleiotropic loci were also mapped to chromosomal genes (Gene_pw_) by using the online web tool Variant Effect Predictor (VEP, https://grch37.ensembl.org/Tools/VEP, accessed on 23 March 2023).

The MAGMA program utilized original p-values from the SNP-based analysis of GWAS to identify candidate genes associated with SA and MDD, respectively. The program aggregates the statistical significance of SNPs within a gene to output a gene-based statistic [40]. Genes with a *p*-value below 0.05 were considered significantly associated with each trait (Gene_magma-SA_, Gene_magma-MDD_). After that, the two gene sets were intersected to obtain the pleiotropic genes for SA and MDD (Gene_magma_).

In the third approach, MetaXcan, the latest implementation program of PrediXcan, integrates genomic information from *cis*-eQTL (expression quantitative trait loci) from GTEx with GWAS summary-level data of 2 traits without requiring individual-level genotyping data. The program was considered as one more applicable way to identify trait-associated genes in specific tissues with the increasing publicly available GWAS summary datasets [41]. MetaXcan mainly contains two programs, S-PrediXcan and S-MultiXcan. Single tissue-associated genes (e.g., whole blood, WHB) were calculated using the S-PrediXcan program. The latter program was applied to compute associations for multiple tissues and integrate measurements across tissues (such as 13 subtypes of brain tissue). The program was able to reduce the false negative rate caused by the strict Bonferroni correction when conducting multiple S-PrediXcan analyses on multiple tissues [42]. Those genes with a *p*-value below 0.05 were considered significantly associated with each trait in specific tissues (Gene_brain-SA_, Gene_whb-SA_; Gene_brain-MDD_, Gene_whb-MDD_). Subsequently, we further took the intersection of two trait-associated genes by the two different tissues (Gene_brain_ and Gene_whb_).

### 2.4. Enrichment Analysis of Pleiotropic Genes for SA and MDD

The Gene Ontology (GO) sets and Kyoto Encyclopedia of Genes and Genomes (KEGG) pathways enrichment analyses were performed for the above four sets of pleiotropic genes (Gene_pw_, Gene_magma_, Gene_brain_ and Gene_whb_) using the R package ‘ClusterProfiler’ to explore the shared molecular mechanism in which pleiotropic genes would play a role in the occurrence of SA and MDD, simultaneously [43]. The enriched GO set or KEGG pathway from the pleiotropic genes with a *p*-value below 0.05 was considered statistically significant.

## 3. Results

### 3.1. Selection of Independent Genetic Instruments

According to the pre-set criteria (*r*^2^ < 0.1, distance = 1000 kb), we first selected suitable independent genetic instruments from the exposure GWAS datasets. In MR analyses that treated SA as an exposure, independent genetic instruments were selected from three groups of independent IVs (Group 1: *p* < 5 × 10^−6^, *n* = 15; Group 2: F > 10, *n* = 569; Group 3: *p* < 5 × 10^−3^, *n* = 1518). If the case of treating MDD as an exposure, we also obtained 7, 1535 and 4024 independent IVs from three groups of independent IVs in the GWAS summary dataset of MDD. The detailed number of independent SNPs used in the subsequent MR analyses is also shown in Table 2.

### 3.2. Genetic Causal Correlations between SA and MDD

In MR analyses, we first treated SA as an exposure and MDD as an outcome with three different sets of independent IVs to explore whether there was a significant causal effect of SA on MDD. By adopting the most suitable MR method, our MR analysis with strong IVs with *p*-value below 5 × 10^−6^ found that SA was a causal risk factor for MDD (fixed-effects IVW, OR = 1.07, *p* = 5.01 × 10^−3^) (Table 2 and Figure 2A), which was also observed in the other three MR methods, including random-effects IVW (OR = 1.07, *p* = 5.01 × 10^−3^), weighted median (OR = 1.07, *p* = 1.76 × 10^−2^), and Robust Adjusted Profile Score (OR = 1.07, *p* = 8.50 × 10^−3^) (Appendix A). In addition, the risk causal effect of SA on MDD was further confirmed in the other two groups of independent IVs (F > 10: OR = 1.04, *p* = 1.21 × 10^−2^; *p* < 5 × 10^−3^: OR = 1.02, *p* = 2.81 × 10^−2^) (Table 2 and Figure 2B,C). Notably, the acting directions of all three sets of MR analyses were indicated to be true in the directionality tests.

In contrast to the foregoing sets, we treated MDD as an exposure and SA as an outcome to evaluate if the occurrence of SA would be affected by MDD. However, we did not find any reverse significant causal effect of MDD on SA with any of the MR methods (*p* > 3.98 × 10^−1^) (Table 2 and Figure 2D,E). The directionality tests also suggested that the causal effect of MDD on SA in the East Asian population may be false.

### 3.3. Identification of a Shared Genetic Basis between SA and MDD

Given the significant causal effect of SA on MDD, we further employed the GWAS-PW program to investigate if there was a shared genetic basis for the two diseases. The program successfully identified seven genetic loci with a PPA3 value greater than 0.8, indicating that they could be able to jointly influence the two diseases (PPA3 = 0.8074~1.00, Table 3). The positive findings also suggested that the two diseases had a shared genetic basis. Subsequently, the VEP program mapped four of the loci into four different genes (*EIF4G3*, *SLC9B1P2*, *HMGCLL1* and *PDE2A*) by chromosomal position, respectively (Table 3). The GO set enrichment analysis found that the four genes were significantly enriched in the gene set of positive regulation of inflammatory response (*p* = 2.42 × 10^−2^) (Figure 3).

### 3.4. Identification of Pleiotropic Genes and Pathways for SA and MDD

To further investigate genetic mechanisms underlying the causal correlations between SA and MDD, we utilized the MAGMA and MetaXcan programs to identify pleiotropic genes and pathways for SA and MDD. At a whole-genome level, the MAGMA program identified 833 and 1088 genes significantly associated with SA and MDD, respectively. The two sets of associated genes were taken as an intersection to extract a group of 44 pleiotropic genes (Gene_magma_) (Table 4), which was nominally enriched in immune-related pathways, such as the antigen processing and presentation pathway (*p* = 1.46 × 10^−2^) (Figure 4A).

At a tissue-specific level, the S-MultiXcan program found a group of 580 genes significantly associated with SA (Gene_brain-SA_) and another group of 303 genes significantly associated with MDD in brain tissue (Gene_brain-MDD_) (Table 4). Seventeen pleiotropic genes could be extracted from the two groups of associated genes (Gene_brain_) and were found to be significantly enriched in the asthma-related pathway (*p* = 3.28 × 10^−2^) (Figure 4B). Enrichment analysis of GO sets suggested that these pleiotropic genes were significantly enriched in immune-related sets, such as alpha-beta T-cell activation (*p* = 7.13 × 10^−3^), NK T-cell differentiation (*p* = 7.38 × 10^−3^), MHC class II protein complex assembly (*p* = 1.18 × 10^−2^), and peptide antigen assembly with MHC class II protein complex (*p* = 1.18 × 10^−2^) (Figure 5). Regarding whole-blood tissue, the S-PrediXcan program identified a group of six pleiotropic genes for SA and MDD (Gene_whb_) from the 305 associated genes specific to SA and 94 associated genes specific to MDD (Table 4). The six pleiotropic genes were significantly enriched in GO sets related to tolerance induction, including T-cell tolerance induction (*p* = 3.96 × 10^−3^), regulation of T-cell tolerance induction (*p* = 3.43 × 10^−3^), tolerance induction (*p* = 8.17 × 10^−3^), and regulation of tolerance induction GO sets (*P* = 5.28 × 10^−3^) (Figure 6).

## 4. Discussion

With the biggest summary datasets of GWAS for SA and MDD in the East Asian population, our two-sample, bidirectional MR study identified a significant causal effect of SA on the occurrence rate of MDD, but we did not find a reverse effect of MDD on the incidence rate of SA. Although several previous observational studies implied a potential correlation between SA and MDD, the outcome was mainly supported by indirect evidence, such as the epidemic coincidence of the two diseases and the higher incidence rate of one disease in specific subjects with another disease [13,14,15]. However, the demonstrated evidence was easily affected by substantial confounding factors and potential reverse causality [17,18]. In contrast, the MR approach is considered similar to a natural RCT study because the alleles in independent genetic variants are randomly distributed, which could overcome the two main issues above. In addition to exploring the direct correlation between SA and MDD, the present study further identified a solid shared genetic basis of the two diseases and found a number of pleiotropic genes at a whole-genome level and a tissue-specific level, which provided a reasonable explanation for the causal effect of SA on MDD.

The present study is the first to investigate the causal correlation between SA and MDD based on the publicly available GWAS summary datasets. So far there has been little research on crustacean allergy in a genome-wide genetic structure, especially on SA worldwide [44]. The GWAS summary statistics of SA were conducted on about 8900 Japanese women and constitute the only public summary dataset specific to SA worldwide [30], which made the current study the first one to investigate the causal correlation between SA and MDD. Subsequently, we further identified seven independent genetic loci jointly affecting the two diseases, indicating they indeed have a shared genetic basis. Although the heritability of the two diseases was estimated to be from moderate to high [17,18,20,21], the associated genes specific to one disease could not explain their shared genetic basis [45,46,47,48,49,50,51].

### 4.1. Mediation Effects of Pleiotropic Genes and Pathways on the Causal Correlation between SA and MDD

With multiple bioinformatics tools, the present study successfully identified four pleiotropic genes (*EIF4G3, SLC9B1P2, HMGCLL1* and *PDE2A*) that jointly influence the two diseases. The *PDE2A* gene, together with *IL13* and *CYFIP2,* was significantly associated with IgE elevation at 3 years of age (*p* = 5.98 × 10^−7^) [52]. At a whole-genome level, we also identified a group of 44 pleiotropic genes that were highly enriched in the antigen processing and presentation pathway. Among them, the *HSPA1L* gene is an inflammatory factor that is reported to be involved in human immune and inflammatory processes [53].

In brain-specific tissue, seventeen pleiotropic genes were identified and significantly enriched in immune-related pathways. The *TNFRSF14* gene encodes a member of the tumor necrosis factor receptor superfamily and is involved in signal transduction pathways that activate an inflammatory and inhibitory T-cell immune response. Upon binding to CD160 on activated CD4^+^ T cells, this gene could down-regulate CD28 costimulatory signaling, restricting memory and the alloantigen-specific immune response [54]. The gene was also found to be involved in two-way cell–cell contact signaling between antigen-presenting cells and lymphocytes [55]. Besides that, the *ITK* gene encodes an intracellular tyrosine kinase expressed in T cells. The ITK signaling is essential for Th2 differentiation and subsequent production of related cytokines, which is thought to play an important role in T-cell proliferation and differentiation [56]. In addition, multiple murine models have demonstrated the importance of ITK in asthma and atopic dermatitis, e.g., the risk of atopic dermatitis was increased with the evaluated level of ITK [57]. The inhibition of ITK therefore may be a useful strategy to treat allergic diseases. Genetic studies also found that the polymorphisms in the promoter region of this gene were significantly associated with an increased risk of allergic asthma [57].

The six pleiotropic genes in whole-blood specific tissue were significantly enriched in T-cell tolerance induction GO sets. Notably, three of these genes (*CEPT1, PHLPP1, USP3*) were reported to directly or indirectly link to inflammatory bowel disease through experimental or bioinformatics methods [58,59,60]. Moreover, the *PHLPP1* gene is involved in the development and function of regulatory T cells. This gene plays a vital role in restricting the innate immune responses of macrophages, and further studies suggested that it may be a potential therapeutic target within the context of dysregulated macrophage activity [61]. The deubiquitination of the *USP3* gene could promote the assembly and activation of inflammatory bodies, which play a vital role in the innate immune system [60]. The *CEPT1* gene is essential for endothelial cell function and tissue recovery after ischemia, and that fenofibrate rescues CEPT1-mediated activation of PPARα [62].

These identified pleiotropic genes were found to be enriched in allergy and immune-related pathways/GO sets, including positive regulation of inflammatory response, antigen processing and presentation pathway, asthma-related pathway, T-cell activation/differentiation GO sets and T-cell tolerance induction GO sets. These pathways/GO sets are well known to be involved in allergic reactions [63]. One previous study observed that psychiatric patients with suicide attempts had an elevated level of IL-1β and IL-6 in blood and brain tissues, suggesting that the occurrence of psychiatric disorders was closely related to the dysfunction of the immune system that is caused by an inflammatory response [64]. Holmes et al. found a higher expression level of translocator protein (TSPO) in MDD patients with suicidal thoughts than those without suicidal thoughts [65]. The TSPO could stimulate microglia in the brain, causing the latter to secrete cytokines that induce inflammation response in the brain. It is reported that more than 80% of 184 adolescent patients who were experiencing their first episode of MDD presented prolonged food intolerance with an elevated level of serum histamine, resulting in hyperpermeability of the blood-brain barrier, and it has been implicated that prolonged high levels of serum histamine could be a risk factor for MDD [66]. Taken together, allergy- and immune-related pathways, especially for inflammatory responses, may plausibly play a vital role in the shared genetic basis of the two diseases and are believed to be key molecular mechanisms underlying their causal correlations.

### 4.2. Limitations

The present study has several kinds of limitations. First, the whole sample size and number of cases for SA in the East Asian population is relatively small, which may affect the reliability of outcomes from GWAS analysis and cause those associated variants with a smaller effect to be difficult to be uncovered. The small number of genotyped SNPs for SA on a custom East Asian-specific Axiom array also limited the overlapping number of variants with the bigger number of genotyped SNPs for MDD, causing one of our MR analyses for the effect of MDD on SA unavailable to conduct. Despite that, our MR analyses based on independent and strong IVs supported the negative effect of MDD on SA. Furthermore, the incidence of SA was self-reported from the participants, which may potentially affect the accuracy of phenotyping for SA and induce self-reporting bias or selective recall. Secondly, the use of GWAS study for SA is very limited up to now. We only collected one summary dataset of GWAS for SA in Japanese from publicly available database worldwide, which make the follow-up validation analysis in an independent GWAS dataset unachievable. However, the identified causal effect of SA on MDD was cross-validated by the pre-set three groups of independent genetic variants. Moreover, the individual genotyping data from the UK Biobank are a good source for performing GWAS analyses on various seafood allergies in Europe, which would enable us to validate the findings in a different population and even perform across-ancestry comparisons. The last but not least point is about the biological role of the pleiotropic genes in the incidence of the two diseases, which should be further investigated by well-designed experimental approaches.

## 5. Conclusions

Our two-sample, bidirectional MR analyses suggested a significant causal effect of SA on the incidence rate of MDD in the East Asian population, but the reverse was not true. In addition, we also identified seven genetic loci that jointly influence the two diseases, indicating that SA and MDD indeed have a shared genetic basis. Enrichment analyses of pleiotropic genes at both whole-genome and tissue-specific levels implied the involvement of allergy and immune-related pathways/GO sets in the shared molecular mechanisms of the two diseases. Elucidating the causal effect and the acting direction may be beneficial in reducing the incidence rate of MDD for the massive group of SA patients in the East Asian region.

## Figures and Tables

**Figure 1 nutrients-16-00079-f001:**
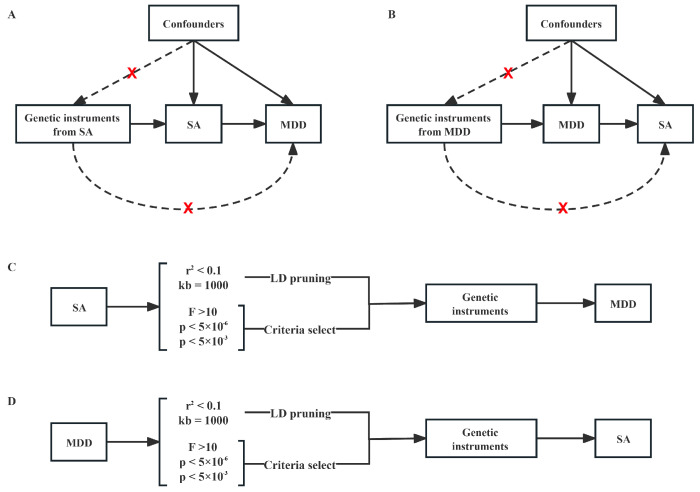
**Schematic diagram of principal MR analyses and main flowchart of this study.** (**A**,**B**) Schematic diagram for exploring bidirectional causal effects between SA and MDD; (**C**,**D**) Schematic diagram for investigating causal effect of SA on MDD or MDD on SA with independent genetic instrumental variables using pre-set criteria.

**Figure 2 nutrients-16-00079-f002:**
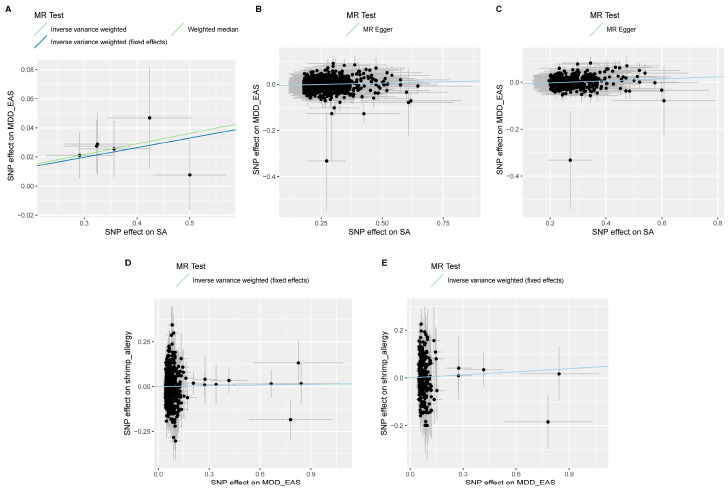
**Scatter plots showing significant causal effects between SA and MDD.** (**A**–**C**) Significant causal effects of SA on MDD with IVs with *p*-value below 5 × 10^−6^, 5 × 10^−3^ and F > 10, respectively; (**D**,**E**) Causal effects of MDD on SA with IVs with *p*-value below 5 × 10^−3^ and F > 10, respectively.

**Figure 3 nutrients-16-00079-f003:**
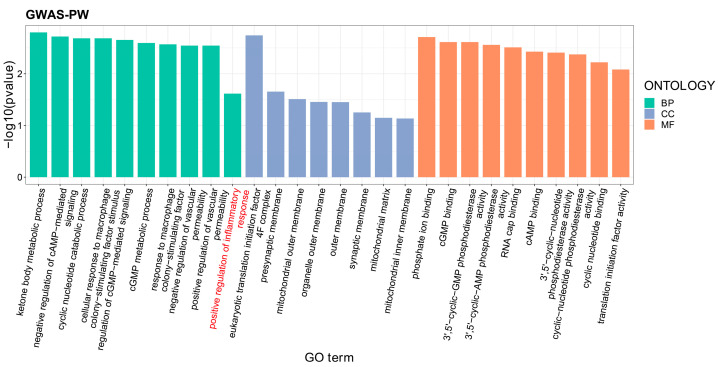
**Enrichment analyses of gene ontology sets for pleiotropic genes of SA and MDD from the GWAS-PW program.** *X*-axis: The GO terms from left to rights belong to biological process, cellular component and molecular function, respectively; *Y*-axis: −log_10_(*p*-value).

**Figure 4 nutrients-16-00079-f004:**
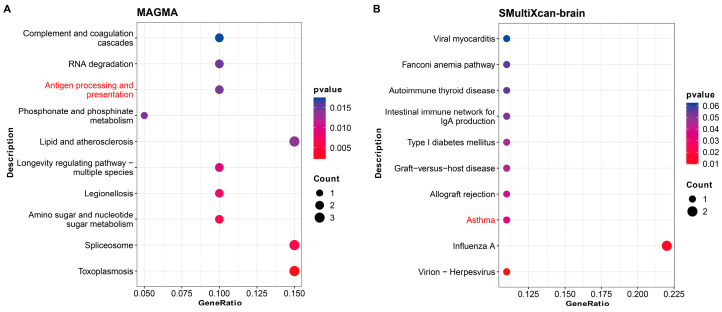
Enrichment analyses of KEGG pathways for pleiotropic genes at a whole-genome level in MAGMA program (**A**) and in brain tissue by the SMultiXcan program (**B**).

**Figure 5 nutrients-16-00079-f005:**
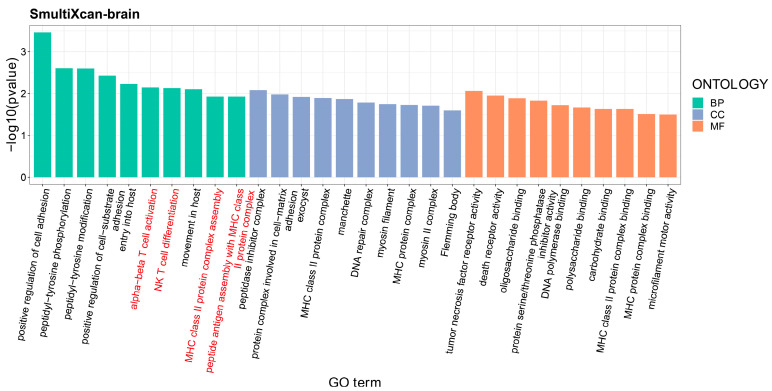
**Enrichment analyses of gene ontology sets for pleiotropic genes in brain tissue by the SMultiXcan program.** *X*-axis: The GO terms from left to right belong to biological process, cellular component, and molecular function, respectively; *Y*-axis: −log_10_(*p*-value).

**Figure 6 nutrients-16-00079-f006:**
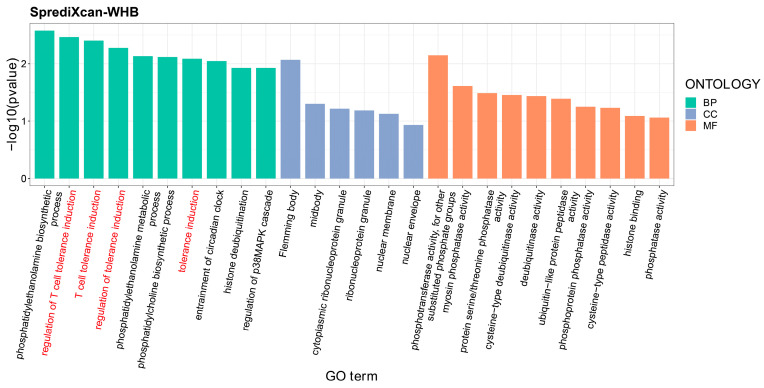
**Enrichment analyses of gene ontology sets for pleiotropic genes in whole blood tissue by the PrediXcan program.** *X*-axis: The GO terms from left to right belong to biological process, cellular component, and molecular function, respectively; *Y*-axis: −log_10_(*p*-value).

**Table 1 nutrients-16-00079-t001:** Basic characteristics of GWAS summary datasets for SA and MDD.

Characters/Phenotype	Shrimp Allergy	Major Depressive Disorder
**Abbreviation**	SA	MDD
**Cases**	539	15,771
**Controls**	8350	178,777
**Original no. of SNPs**	581,673	7,440,922
**No. of processed SNPs**	570,779	5,437,803
**Overlapped SNPs**	494,741	
**Ethnicity**	East Asian	

Overlapped SNPs: Overlapped SNPs between GWAS datasets of SA and MDD.

**Table 2 nutrients-16-00079-t002:** Overall MR analyses with selected genetic instruments for causal correlations between SA and MDD.

Exposure	Outcome	No. of Clumped SNPs ^a^	No. of SNPs in MRA ^b^	MR *p*-Value ^c^	MR Method ^d^	Heterogeneity	Pleiotropy	Directionality
*Independent genetic SNPs with p < 5 × 10^−6^ (Group 1)*								
SA	MDD	15	6	**5.01 × 10^−3^**	fixed-effects IVW	No	No	TRUE
MDD	SA	7	0	ND ^e^	--	--	--	--
*Independent genetic SNPs with F > 10 (Group 2)*								
SA	MDD	569	473	**1.21 × 10^−2^**	MR Egger	No	Yes	TRUE
MDD	SA	1535	265	3.98 × 10^−1^	fixed-effects IVW	No	No	FALSE
*Independent genetic SNPs with p < 5 × 10^−3^ (Group 3)*								
SA	MDD	1518	1302	**2.81 × 10^−2^**	MR Egger	No	Yes	TRUE
MDD	SA	4024	690	6.87 × 10^−1^	fixed-effects IVW	No	No	FALSE

^a^ No. of clumped SNPs: number of independent genetic SNPs after clumping; ^b^ No. of SNPs in MRA: number of independent genetic SNPs used in the MR analysis for each pair of exposure and outcome; ^c^ MR *p*-value: *p*-value of the most suitable MR method; ^d^ MR Method: the most suitable MR analysis used in MR analysis; ^e^ ND: No data were obtained as there is no overlapped SNP after clumping.

**Table 3 nutrients-16-00079-t003:** Shared genetic loci and mapped genes from GWAS-PW program.

SNP ID	Chromosome	Physical Position	PPA3	Mapped Gene
rs76337386	chr1	21493585	0.8174	*EIF4G3*
rs17130426	chr1	88790511	0.8195	-
rs1281337	chr1	182111988	0.8913	-
rs118061427	chr2	92108070	1	*SLC9B1P2*
rs4236131	chr6	55319676	0.8074	*HMGCLL1*
rs341058	chr11	72385725	0.8581	*PDE2A*
rs16954906	chr13	98747021	0.8328	-

PPA3: posterior probabilities of association in model 3.

**Table 4 nutrients-16-00079-t004:** Pleiotropic genes of SA and MDD identified at a whole-genome level using the MAGMA program and a tissue-specific level by the MetaXcan program.

Methods	GWAS Dataset	N	*p* < 0.05	Overlapped Genes
MAGMA	SA (Whole-genome)	16,644	833	44 (*ABHD2*, *AFF3*, *C2*, *CCDC83*, *CCDC88B*, *CFB*, *CHPT1*, *DARS1*, *ECE1*, *FAM181B*, *FKBP1A*, *GFI1B*, *GNPDA2*, *GPR18*, *GPR20*, *HSPA13*, *HSPA1A*, *HSPA1L*, *ID4*, *IFT122*, *KIF3C*, *LSM2*, *LYPD6*, *MAP2K3*, *MBD4*, *NFIX*, *PEAK1*, *PGM2*, *PPP4R4*, *PRDM14*, *PTCH1*, *PTP4A3*, *R3HDML*, *RASAL1*, *ROPN1L*, *SEMA3D*, *SERPINA10*, *SH3TC1*, *SKIC2*, *TES*, *THBS2*, *TLE3*, *TMEM25*, *TTC36*)
	MDD (Whole-genome)	18,162	1088	
MetaXcan	SA (brain)	12,197	580	17 (*C2orf42*, *EXOC2*, *FANCI*, *HLA-DMA*, *IQCG*, *ITK*, *ITLN1*, *LCE1C*, *LINC00581*, *MYH7B*, *PPP1R14B*, *RAMP3*, *RNF175*, *RSU1*, *TNFRSF14*, *TPSD1*, *VTN*)
	MDD (brain)	12,258	303	
	SA (whole blood)	6383	305	6 (*CEPT1*, *GNB1L*, *PHLPP1*, *USP3*, *ZNF440*, *ZNF670*)
	MDD (whole blood)	7206	94	

N: number of genes mapped at a whole-genome level or a tissue-specific level.

## Data Availability

Publicly available datasets were analyzed in this study. These datasets can be found here: https://www.ncbi.nlm.nih.gov/pmc/articles/PMC5773682/#MOESM4 (accessed on 13 December 2023) and https://figshare.com/articles/dataset/mdd2021asi/16989442 (accessed on 13 December 2023).

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
