# Peer review of "A Positive Causal Effect of Shrimp Allergy on Major Depressive Disorder Mediated by Allergy- and Immune-Related Pathways in the East Asian Population"

_nutrients, 2023, doi:10.3390/nu16010079_

Round 1

Reviewer 1 Report

Comments and Suggestions for Authors

1. Observational studies with this type of correlation are very complex, but innovative and of global interest, there are many mechanisms of fesa and cell interaction to study and depend on each region.

2.The genetic study of this East Asian population must also compare the American population, for example, and also the South American population, where there are different types of curstaceans.

3. Have you studied any interactions with the intestinal microbial system?

4. Were these enrichment analyzes of pleiotropic genes at specific genome levels in specific tissues? Nervous system and blood for example?

5. The immune system is a self-recognition system and this genetic mechanism can confuse diseases.

6. Elucidating the causal effect and direction of action may be beneficial in reducing the incidence rate of depressive disorders for a huge group of patients with these allergies in the East Asian region.

Reviewer 2 Report

Comments and Suggestions for Authors

Line 61, The two diseases are different, make it coherent to introduce MDD in the paragraph. 

Line 66, Modify the sentence. According to Allergen Online, there are 14 major and, 8 common allergens in the EU.

Lines 93-100. This section should be in the methodology. It would be better to add the aims of the study in the last paragraph of the introduction.

All figures data missing. It's hard to analyze the text without the results marked in the figure.

Line 188, add space before 0.8

Comments on the Quality of English Language

Overall presentation of the paper is good except that all figures are missing. Without the results marked in the figures, the review is incomplete.

Reviewer 3 Report

Comments and Suggestions for Authors

This is one of the few studies that uses Mendelian Randomization (MR) analysis on a large GWAS dataset to assess if there is a causal relationship between shrimp allergy (SA) and major depressive disorder (MDD). GO sets and KEGG pathways were also analyzed to assess possible genetic mechanisms. The study shows that 7 pleiotropic gene variants were shared between SA and MDD. Further, enrichment analysis showed the involvement of 17 brain-specific pleiotropic genes and six whole-blood tissue-related genes that regulated immunity and allergy-related pathways. The study shows a significant causal association between SA and MDD, in that SA may be a causative risk factor for MDD, but not vice versa. It is a novel and well-written study and opens many avenues for further research, however at its initial stage, the findings are still to be validated experimentally to be considered causal.

1.       Line 118-119; 128-130:  The SA population involved in the study is restricted to the Japan region, whereas the MDD population is derived from China, Taiwan, and East Asian populations from the UK and USA. The comparison with the Japanese population with SA, with the rest of the East Asian population may not represent a complete comparison. The authors should consider adding the rationale for the comparison in the study and also discuss databases where comparisons can be made in similar populations.

2.       Line 118: The study uses self-reported SA incidence, which has its own limitations like self-reporting bias or selective recall. The authors should consider including this as a study limitation.

3.       Authors can discuss the biological roles of the identified genes and how they can be relevant to induce SA or MDD phenotype.

4.       A table with commonly used abbreviations would be helpful for the readers.
